# Environmental changes in oxygen tension reveal ROS-dependent neurogenesis and regeneration in the adult newt brain

L Shahul Hameed[1], Daniel A Berg[1†], Laure Belnoue[1], Lasse D Jensen[2,3], Yihai Cao[2,3,4], András Simon[1]*

[1]Department of Cell and Molecular Biology, Karolinska Institute, Stockholm, Sweden; [2]Department of Microbiology, Tumor and Cell Biology, Karolinska Institute, Stockholm, Sweden; [3]Department of Medical and Health Sciences, Linköping University, Linköping, Sweden; [4]Department of Cardiovascular Sciences, NIHR Leicester Cardiovascular Biomedical Research Unit, Glenfield Hospital, University of Leicester, Leicester, United Kingdom

**Abstract** Organisms need to adapt to the ecological constraints in their habitat. How specific processes reflect such adaptations are difficult to model experimentally. We tested whether environmental shifts in oxygen tension lead to events in the adult newt brain that share features with processes occurring during neuronal regeneration under normoxia. By experimental simulation of varying oxygen concentrations, we show that hypoxia followed by re-oxygenation lead to neuronal death and hallmarks of an injury response, including activation of neural stem cells ultimately leading to neurogenesis. Neural stem cells accumulate reactive oxygen species (ROS) during re-oxygenation and inhibition of ROS biosynthesis counteracts their proliferation as well as neurogenesis. Importantly, regeneration of dopamine neurons under normoxia also depends on ROS-production. These data demonstrate a role for ROS-production in neurogenesis in newts and suggest that this role may have been recruited to the capacity to replace lost neurons in the brain of an adult vertebrate.

*For correspondence: Andras. Simon@ki.se

**Present address:** †School of Medicine, Johns Hopkins University, Baltimore, United States

**Competing interests:** The authors declare that no competing interests exist.

## Introduction

Animals that experience episodes of low oxygen concentration use different strategies to protect their organs, particularly those that are metabolically highly active, such as heart and brain. These species are in general capable of adjusting their metabolic rate and to cope with the accumulation of anaerobic by-products (*Larson et al., 2014*). Re-oxygenation upon return to normoxia may lead to the production of harmful reactive oxygen species (ROS), and it has been proposed that these animals need to repair tissues that might become damaged during re-oxygenation (*Bickler and Buck, 2007*). In this article, we set out to test this latter hypothesis in the highly regenerative aquatic salamander, the red-spotted newt. We specifically asked whether hypoxia and re-oxygenation leads to events in the newt brain that share common features with processes taking place after experimental ablation and subsequent regeneration of neurons under normoxia.

Adult red-spotted newts are post-metamorphic amphibians with lungs. They remain active all year round and can be found in deep water under ice during winter—an environment known to become hypoxic (*Berner and Puckett, 2010*). The red-spotted newt also possesses a wide spectrum of abilities of regenerating complex structures, including the central nervous system (*Parish et al., 2007*; *Brockes and Kumar 2008*). Of particular importance in the context of the present study is their capacity to replace specific neuronal subtypes in the brain following chemical ablation (*Parish et al., 2007*;

**eLife digest** During the winter, red-spotted newts remain active in water that is covered by ice. The oxygen levels under the ice tend to drop and so the newts adjust their metabolism to cope with these conditions. However, when oxygen levels return to normal, this may result in the newts producing larger amounts of chemically reactive molecules called reactive oxygen species (ROS). These molecules form naturally as a by-product of oxygen metabolism, but in high quantities they can damage cells and tissues.

It has been proposed that red-spotted newts and other animals that experience periods of low oxygen may have evolved processes to repair such damage. Unlike us, red-spotted newts are able to replace nerve cells in the brain that have died or been injured. This regeneration is fuelled by stem cells called ependymoglia cells, which divide to produce new nerve cells. Here, Hameed et al. investigated whether the return of oxygen to normal levels after a period of low oxygen can damage nerve cells in the newts, and whether this is followed by regeneration.

The experiments show that nerve cells in the newt brain do indeed die when oxygen levels return to normal. Also, the brain activates an injury response that triggers the ependymoglia cells to divide. During this process, the ependymoglia cells accumulate ROS and their ability to divide is impaired if the production of ROS is blocked. The replacement of injured brain cells in normal oxygen conditions also depends on increased ROS levels.

Together, Hameed et al.'s findings demonstrate a key role for ROS production in controlling the regeneration of damaged nerve cells in the red-spotted newt. A future challenge is to identify the genes that control the survival and activation of ependymoglia cells in response to increased ROS levels in the brain.

*Berg et al., 2010*). Neuronal regeneration leads to complete restoration of the original status in terms of functional recovery and in the terms of reaching the correct number of neurons in all brain regions tested so far (*Berg et al., 2011*).

Regeneration of neurons is fuelled by the activation and subsequent neurogenesis by neural stem cells (NSCs), the so-called ependymoglia cells (*Berg et al., 2010*; *Kirkham et al., 2014*). Ependymoglia cells line the brain ventricles, express the intermediate filament protein GFAP (glial fibrillary acidic protein), and have radial extensions, reaching the pial surface (*Parish et al., 2007*). Regeneration in homeostatically non-germinal niches is independent of the normal constitutive neurogenesis occurring in the forebrain (*Kirkham et al., 2014*), thus the newt brain is an ideal model for studying both constitutive and injury-induced adult neurogenesis, as well as the relationship between the two.

In order to test whether environmental shifts in oxygen tension lead to events in the adult newt brain that share features with processes taking place during neuronal regeneration under normoxia, we carried out studies on neurogenesis both during shifting and normal oxygen tension. We find that modulation of oxygen tension leads to loss of neurons, activation of microglia, accumulation of ROS in ependymoglia cells concomitant with their cell cycle reentry, and increased neurogenesis in the forebrain. Inhibition of microglia activation does not abolish ependymoglia activation upon re-oxygenation, and NSCs cultured as neurospheres respond by increased proliferation in vitro, both observations indicating a cell autonomous role for ROS in NSCs. We further show that ROS production is required for cell cycle reentry by ependymoglia cells as well as for neuronal regeneration in the normally quiescent midbrain also during normoxia. Thus, we show that ROS production is an important component of NSC regulation and propose that this role of ROS may have been recruited during evolution to the capacity of regenerating neuronal subpopulations.

## Results

### Hypoxia followed by re-oxygenation induces neuronal death and an inflammatory response

First, we tested whether red-spotted newts were able to cope with hypoxia. We placed animals into an aquarium sealed with a plastic lid in which the oxygen level was manipulated by perfusing the water

with nitrogen gas. The oxygen concentration was monitored by electrodes, which were feeding back to a unit controlling the gas supply to the water (*Figure 1—figure supplement 1A*). We tested various regimens of shifting oxygen tension and found that newts were able to cope with hypoxic conditions as low as 10% of the normal oxygen tension provided that the decrease was gradual over a period of 48 hr. Hence, animals were kept in 10% of the normal oxygen tension for five days, and brought subsequently back instantly to normoxic conditions, and analyzed at different time points (*Figure 1—figure supplement 1B*).

In order to determine whether hypoxia and subsequent re-oxygenation causes injury to the brain, we performed TUNEL (Terminal deoxynucleotidyl transferase dUTP nick end labeling) staining, which identifies cells in the late phase of apoptosis (*Zhao et al., 2001*; *Arama and Steller, 2006*). We found a 2.0-fold increase in the number of TUNEL$^+$ cells after 5 days of hypoxia in the forebrain. The number of apoptotic cells was further increased to 3.3-fold of the normal, after 1 day of re-oxygenation (*Figure 1A,B*). To determine whether neurons are lost during hypoxia/re-oxygenation, we carried out double immunostaining for the pan-neuronal marker NeuN and TUNEL. We found that the number of neurons with apoptotic phenotype was elevated showing a 3.2-fold increase after re-oxygenation compared to control animals (*Figure 1C*). These results indicated that hypoxia/re-oxygenation leads to neuronal loss in the newt brain.

Activation of microglial cells is a hallmark of an injury response in the vertebrate brain (*Gonzalez-Scarano and Baltuch, 1999*). We have previously reported that microglial cells expressing IBA1 (ionized calcium-binding adaptor molecule 1) become activated following selective, toxin-mediated ablation of neuronal subpopulations in the newt brain (*Kirkham et al., 2011*). Hence, we next tested whether hypoxia/re-oxygenation leads to a microglia response by assessing the number of proliferating IBA1$^+$ cells in the brain. We found that the number of proliferating IBA1$^+$ cells increased 2.8-fold by hypoxia/re-oxygenation (*Figure 1D,E*). These results collectively show that hypoxia/re-oxygenation leads to loss of neurons in the newt brain and activation of a microglia response.

## Re-oxygenation leads to cell cycle reentry by forebrain ependymoglia cells and increased neurogenesis

Ependymoglia cells give rise to neurons both during homeostasis and following loss of neurons (*Parish et al., 2007*; *Berg et al., 2010*; *Kirkham et al., 2014*) and we next asked whether hypoxia/re-oxygenation leads to increased ependymoglia cell proliferation in the forebrain. We did not find any statistically significant changes in the number of proliferating ependymoglia cells immediately after hypoxia (*Figure 2B*). In contrast, after hypoxia followed by re-oxygenation, we observed a 1.8-fold increase in the number of proliferating ependymoglia cells as assessed by double immunostaining with antibodies against PCNA (proliferating cell nuclear antigen) and GFAP (*Figure 2A,C*). In order to corroborate these observations, we also carried out pulse labeling with the nucleotide analogue EdU, which incorporates into the DNA during S-phase. Animals were injected with EdU 2 hr before sacrificing them. In accordance with the conclusion based on PCNA staining, we observed a 1.8-fold increase in the number of proliferating ependymoglia cells (*Figure 2—figure supplement 1*).

Next, we wanted to test whether the loss of neurons and concomitant injury response also led to increased neurogenesis. We pulsed animals with EdU for five days following re-oxygenation. After 35 days of chase, we detected that the number of newly formed neurons increased by 2.4-fold compared to the control as assessed by the number of cells that had incorporated EdU and were expressing the pan-neuronal marker Hu (*Figure 2D,E*). Hence, we conclude that in addition to loss of neurons, hypoxia/re-oxygenation also leads to increased neurogenesis.

## Ependymoglia cell proliferation and neurogenesis following re-oxygenation depends on ROS production

It has been hypothesized that hypoxia followed by re-oxygenation leads to production of ROS, which on one hand may lead to tissue damage and on the other hand act as signaling molecule (*Li and Jackson, 2002*; *D'Autreaux and Toledano, 2007*). To address whether ROS is produced during hypoxia/re-oxygenation, we incubated brains with the super oxide sensitive dye, hydroethidine (HEt), which upon oxidation produces red fluorescence. We observed increased HEt signal already 6 hr of re-oxygenation, which remained persistent over several days. As illustrated in *Figure 3A*, HEt signal shows ROS accumulation in brain sections, with marked enrichment in ependymoglia cells three days

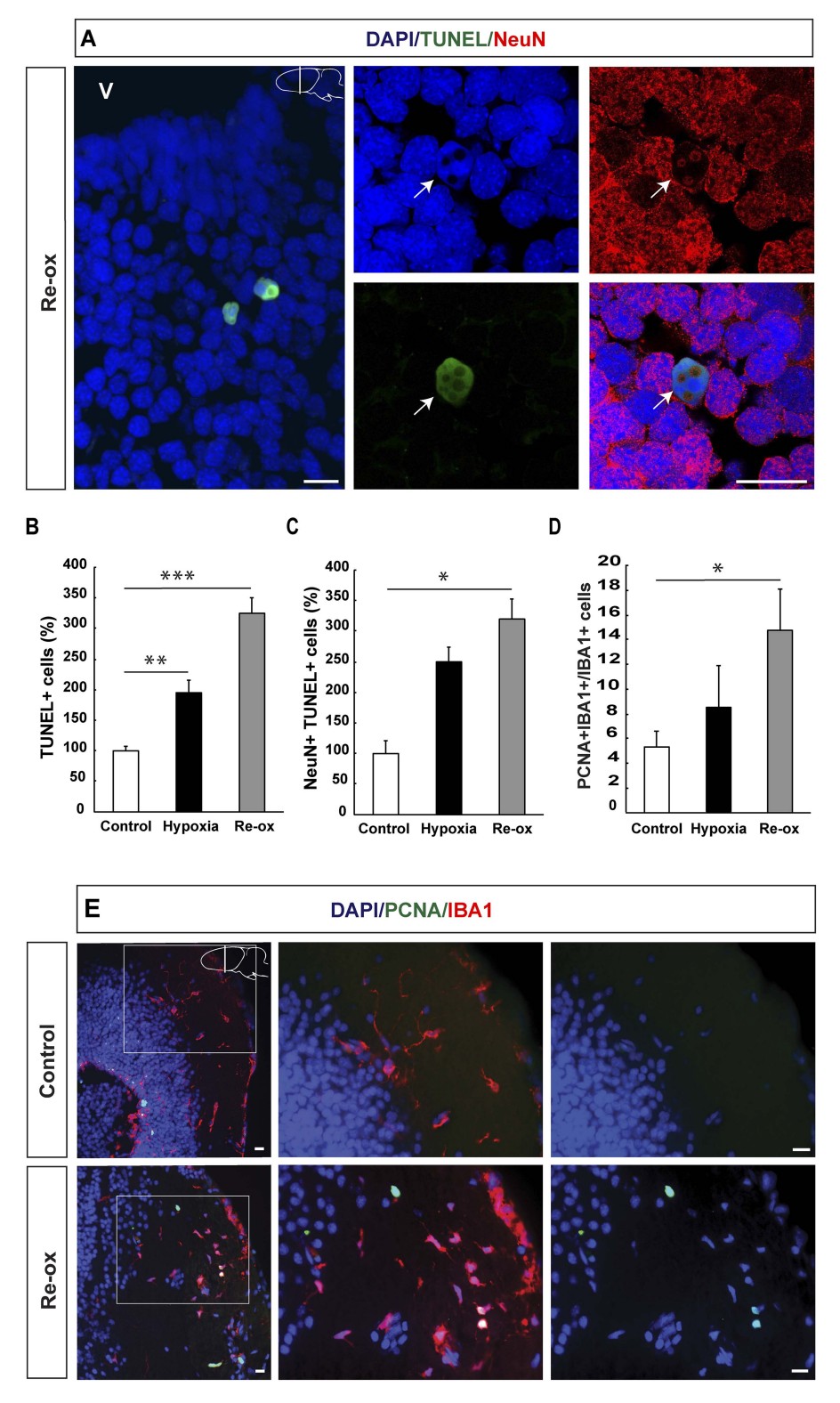

**Figure 1**. Hypoxia/re-oxygenation-induced neuronal cell death and microglia response. (**A**) TUNEL+ cells are shown in the forebrain parenchyma at low magnification. The high-magnification image shows TUNEL+/NeuN+ nucleus (arrow). Note the disappearing NeuN staining in the TUNEL+ cell. (**B**) Quantification of TUNEL+ cells after hypoxia and hypoxia/re-oxygenation. n = 4, **p < 0.01, ***p < 0.001. (Unpaired t-test). (**C**) Quantification of NeuN+/TUNEL+
*Figure 1. continued on next page*

*Figure 1. Continued*

cells after hypoxia and hypoxia/re-oxygenation. n = 4, *p < 0.05. (Mann–Whitney test). (**D**) Quantification of microglia activation after hypoxia and hypoxia/re-oxygenation. n = 4–5, *p < 0.05. (Unpaired t-test). (**E**) Low-magnification image illustrating microglia proliferation in control and experimental animals. Scale bar = 20 µm.

The following source data and figure supplement are available for figure 1:

**Source data 1**. Hypoxia/re-oxygenation-induced neuronal cell death and microglia response.

**Figure supplement 1**. Experimental system for manipulation of oxygen tension.

---

after re-oxygenation. Similar results were obtained when HEt was injected intravenously 60 min prior to sacrificing the animals (data not shown).

Next, we aimed to address the relationship between ROS and ependymoglia cell proliferation. Following re-oxygenation, we injected animals with apocynin, which is an inhibitor of ROS production by interfering with the NADPH oxidase complex (NOX) (*Muijsers et al., 2000*). First, we observed that twice daily apocynin injection over a period of 3 days led to reduced ROS levels as detected by HEt staining (*Figure 3A*). Double immunostaining with PCNA and GFAP revealed that apocynin on its own did not reduce cell proliferation during normoxic conditions (*Figure 3B*). Importantly, after re-oxygenation, the number of proliferating ependymoglia cells was reduced by 2.0-fold (*Figure 3C,D*) in apocynin-injected compared to vehicle-injected animals.

In addition to NOX, ROS may also be generated by the mitochondria (*Prozorovski et al., 2015*) and we first tested whether mitochondrial ROS increased during re-oxygenation. To detect mitochondrial ROS, we incubated newt brains with mitochondrial superoxide indicator Mitosox (*Robinson et al., 2008*). We observed excellent co-localization of the signal with the mitochondrium marker mitofusin-1 (*Rojo et al., 2002*) (*Figure 3E*). Animals that were undergoing hypoxia/ re-oxygenation displayed accumulation of mitochondrial ROS compared to control animals, which could be blocked with administration of the mitochondrially targeted antioxidant, Mitotempo (*Dikalova et al., 2010*) (*Figure 3E*). However, in contrast to the effect of NOX-inhibition with apocynin, Mitotempo administration did not reduce the number of PCNA$^+$ ependymoglia cells (*Figure 3F*).

To address whether inhibition of ROS production ultimately interfered with neurogenesis, we carried out pulse/chase experiments with EdU. Animals were pulsed with EdU during 5 days following re-oxygenation and injected with apocynin for twice per day over a period of 8 days starting directly after re-oxygenation. Following a 35-day chase, we found that the number of EdU$^+$/Hu$^+$ cells was reduced by 2.0-fold compared to vehicle-injected animals (*Figure 3G,H*). These results show that the neurogenic response to hypoxia/re-oxygenation induced neuronal loss depends on ROS production.

Previous studies indicated increased ROS production in newborn neurons (*Tsatmali et al., 2006*), and we aimed to quantify ROS in newborn neurons in the newt brain but this was not feasible due to technical reasons. As an alternative, we used an in vitro culture system. Newt ependymoglia cells form neurospheres under appropriate conditions as described earlier (*Kirkham et al., 2014*). Sphere growth occurs over time as the cells within proliferate, and shifting cells to growth factor free media induces differentiation. In such cultures, we compared the intensity of the ROS indicators HEt in GFAP$^+$ cells and in cells expressing the early neuronal marker doublecortin (DCX). These analyses did not show any difference in signal intensity indicating that ependymoglia cells and young neurons do not differ in terms of ROS production. (*Figure 3—figure supplement 1*).

## Ependymoglia cell proliferation after re-oxygenation is independent of microglia activation

Previous studies in zebrafish have demonstrated a critical role of inflammatory cells, such as microglia, in NSC activation following traumatic brain injury (*Kyritsis et al., 2012*). We next addressed whether microglia activation plays an important role in ependymoglia cell proliferation following hypoxia/ re-oxygenation in the newt. To suppress microglia activation, we administered dexamethasone to animals twice daily for 5 days prior to shifting them to hypoxia and twice daily for three days post re-oxygenation. In accordance with previous reports (*Kirkham et al., 2011*; *Kyritsis et al., 2012*), the

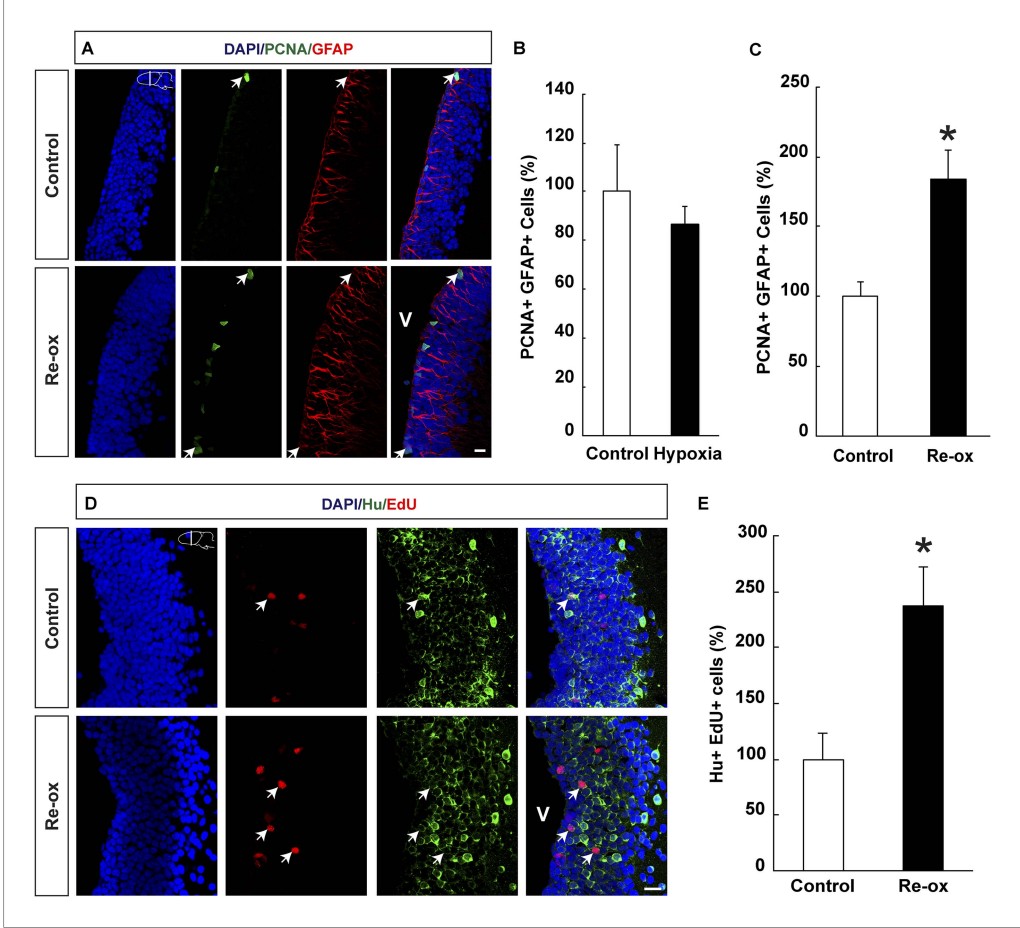

**Figure 2**. Hypoxia/re-oxygenation-induced ependymoglia activation and neurogenesis. (**A**) Low-magnification images illustrating increased proliferation of ependymoglia cells after re-oxygenation. Arrows point to PCNA[+]/GFAP[+] cells. (**B**, **C**) Quantification of PCNA[+]/GFAP[+] ependymoglia cells showing increased ependymoglia proliferation after re-oxygenation but not after hypoxia. n = 4–5, *p < 0.05. (Unpaired t-test for B and Mann–Whitney test for C). (**D**) Low-magnification images illustrating increased number of EdU[+]/Hu[+] cells in the forebrain parenchyma after re-oxygenation. Arrows point to EdU[+]/Hu[+] cells. (**E**) Quantification of EdU[+]/Hu[+] cells indicating increased neurogenesis in the forebrain parenchyma after re-oxygenation n = 5, *p < 0.05. (Unpaired t-test). Scale bar = 20 μm.

The following source data and figure supplement are available for figure 2:

**Source data 1**. Hypoxia/re-oxygenation-induced ependymoglia activation and neurogenesis.

**Source data 2**. Re-oxygenation leads to increased proliferation assessed by EdU incorporation.

**Figure supplement 1**. Re-oxygenation leads to increased proliferation assessed by EdU incorporation.

number of proliferating microglial cells was reduced 3.8-fold compared to vehicle-injected animals (*Figure 4A,B*). In contrast, the proliferative response by ependymoglia cells to hypoxia/re-oxygenation was not altered (*Figure 4C,D*). Consistently, we did not observe any significant decrease of microglia activation in apocynin-treated animals compared to the controls (*Figure 4—figure supplement 1A*). These results indicated that activation of ependymoglia cells was independent of microglia activation following hypoxia/re-oxygenation.

In order to corroborate these findings, we studied the effect on hypoxia/re-oxygenation in neurosphere cultures. When we placed neurospheres into 1% hypoxic chamber for 24 hr and shifted subsequently back to normoxic conditions, we observed a 1.4-fold increase in the number of

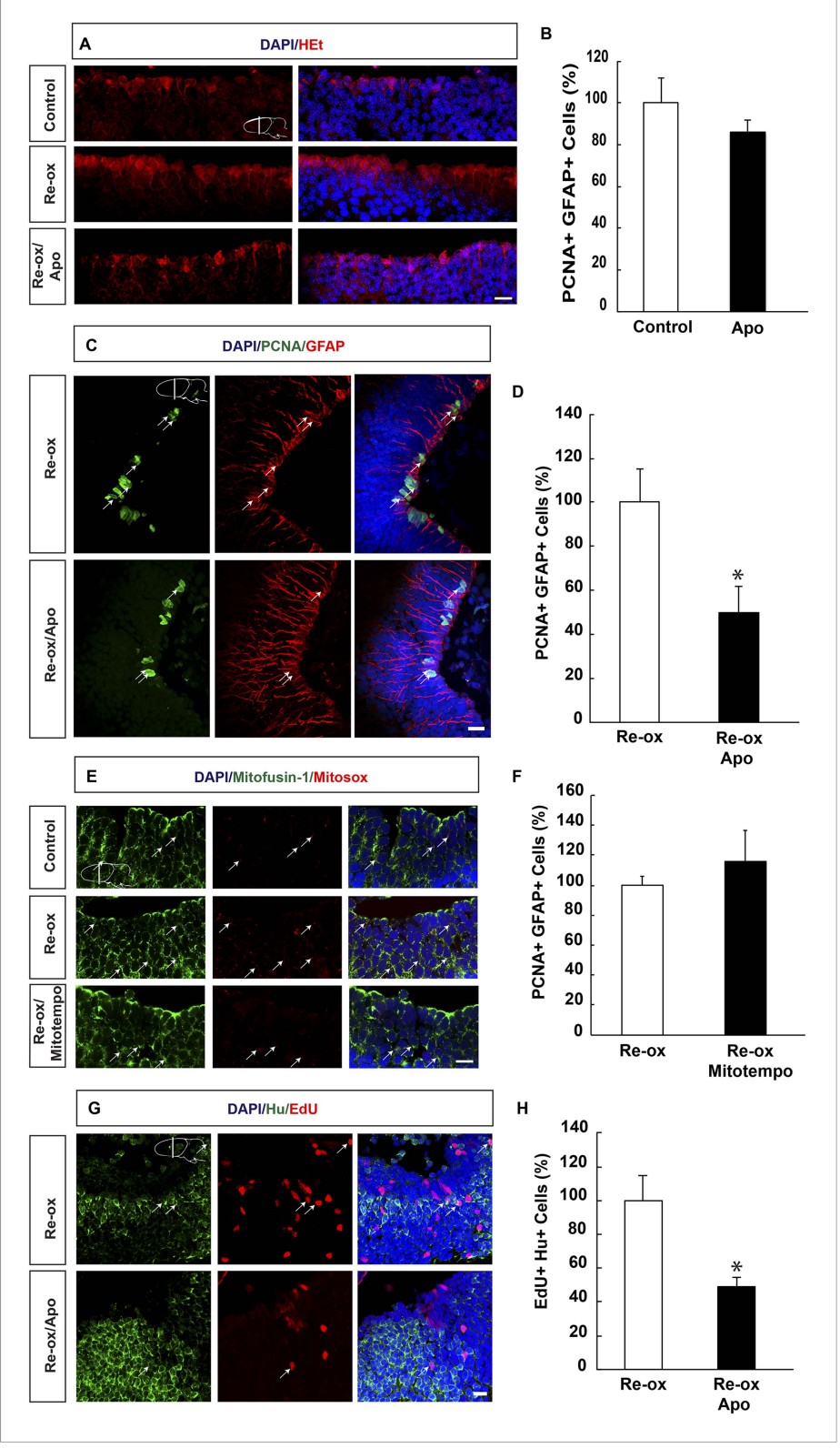

**Figure 3**. ROS-dependent ependymoglia proliferation and neurogenesis. (**A**) Hydroethidine (HEt) shows increased reactive oxygen species (ROS) levels after re-oxygenation particularly in ependymoglia cells. Apocynin inhibits ROS accumulation. (**B**) Quantification of ventricular PCNA+/GFAP+ cells showing that apocynin does not inhibit homeostatic ependymoglia proliferation. n = 4. (Unpaired t-test). (**C**) Low-magnification images illustrating that
*Figure 3. continued on next page*

*Figure 3. Continued*

apocynin decreases the hypoxia/re-oxygenation-induced ependymoglia cell proliferation. Arrows point to PCNA$^+$/GFAP$^+$ cells. (**D**) Quantification of ventricular PCNA$^+$/GFAP$^+$ cells showing that apocynin decreases the hypoxia/re-oxygenation induced ependymoglia cell proliferation. n = 4, *p < 0.05. (Unpaired t-test). (**E**) Images illustrating accumulation of mitochondrial ROS as indicated by Mitosox signal after re-oxygenation. Administration of the mitochondrially targeted antioxidant, Mitotempo reduces mitochondrial ROS. Note the co-localization of Mitosox signal with the mitochondrial marker mitofusin-1. Arrows point to mitofusin-1$^+$/mitosox$^+$ cells. (**F**) Administration of Mitotempo does not change ependymoglia proliferation. n = 4 (Unpaired t-test). (**G**) Low-magnification images illustrating that apocynin decreases the hypoxia/re-oxygenation induced neurogenesis in the forebrain parenchyma. Arrows point to EdU$^+$/Hu$^+$ cells. (**H**) Quantification of EdU$^+$/Hu$^+$ cells showing that apocynin decreases the hypoxia/re-oxygenation induced neurogenesis in the forebrain parenchyma. n = 4–5, *p < 0.05. (Unpaired t-test). Scale bar = 20 μm.

The following source data and figure supplement are available for figure 3:

**Source data 1**. ROS-dependent ependymoglia proliferation and neurogenesis.

**Source data 2**. ROS detection in GFAP$^+$ and DCX$^+$ cells in vitro.

**Figure supplement 1**. ROS detection in differentiating neurons in vitro.

---

proliferating GFAP$^+$ cells in the neurospheres compared to control cultures. This increase was abolished in apocynin-treated cultures indicating a ROS-production dependent response (*Figure 4E,F*). Apocynin on its own did not reduce proliferation (*Figure 4G*). In accordance to these observations, we observed an increase in the number of spheres being formed following re-oxygenation and that this effect was abolished by apocynin treatment (*Figure 4—figure supplement 1B,C*).

Collectively, the above results suggest that the increase in NSC proliferation following re-oxygenation was cell autonomous.

## ROS-dependent regeneration of midbrain dopamine neurons under normoxia

Next, we asked whether regeneration of neurons is dependent on ROS production also during normoxic conditions. To do so, we ablated midbrain dopamine neurons by injecting the neurotoxin, 6-OHDA (6-hydroxydopamine), without applying the hypoxia/re-oxygenation protocol. We previously showed that, in contrast to the forebrain, the newt midbrain is non-germinal and essentially quiescent (*Parish et al., 2007*). However, after administration of 6-OHDA, which kills dopamine neurons in the midbrain within 3 days after injection, newts regenerate lost midbrain dopamine neurons within four weeks. Regeneration is fuelled by the local activation of normally non-proliferating ependymoglia cells, which subsequently undergo a neurogenic program (*Parish et al., 2007*; *Berg et al., 2010*). First, we noticed that injection of 6-OHDA led to accumulation of ROS in ependymoglia cells (*Figure 5A*). Next, we tested whether inhibition of ROS biosynthesis during the regeneration phase interfered with the ablation-responsive cell cycle reentry by ependymoglia cells, by treating animals with apocynin for five days starting from day 4 post-ablation. We found a 3.3-fold reduction in the number of proliferating ependymoglia cells in apocynin- vs control-injected animals (*Figure 5B,C*). In addition, while apocynin treatment alone did not reduce the number of TH (Tyrosine Hydroxylase)-expressing neurons in sham-lesioned control (*Figure 5F*), apocynin inhibited regeneration of dopamine neurons after 6-OHDA-injection assessed by a reduction in the number of midbrain neurons expressing TH (*Figure 5D,E*). These data show that injury responsive ependymoglia proliferation and neuronal regeneration depends on ROS production also during normoxia.

## Discussion

Previous reports provided evidence that ROS signaling and the redox state influence stem cell fate. However, conflicting data exist as to whether ROS impairs or contributes to normal stem cell function (*Kim and Wong, 2009*; *Chuikov et al., 2010*; *Dickinson et al., 2011*; *Le Belle et al., 2011*; *Walton et al., 2012*; *Wang et al., 2013*). We used systemic manipulation of oxygen tension as a means to

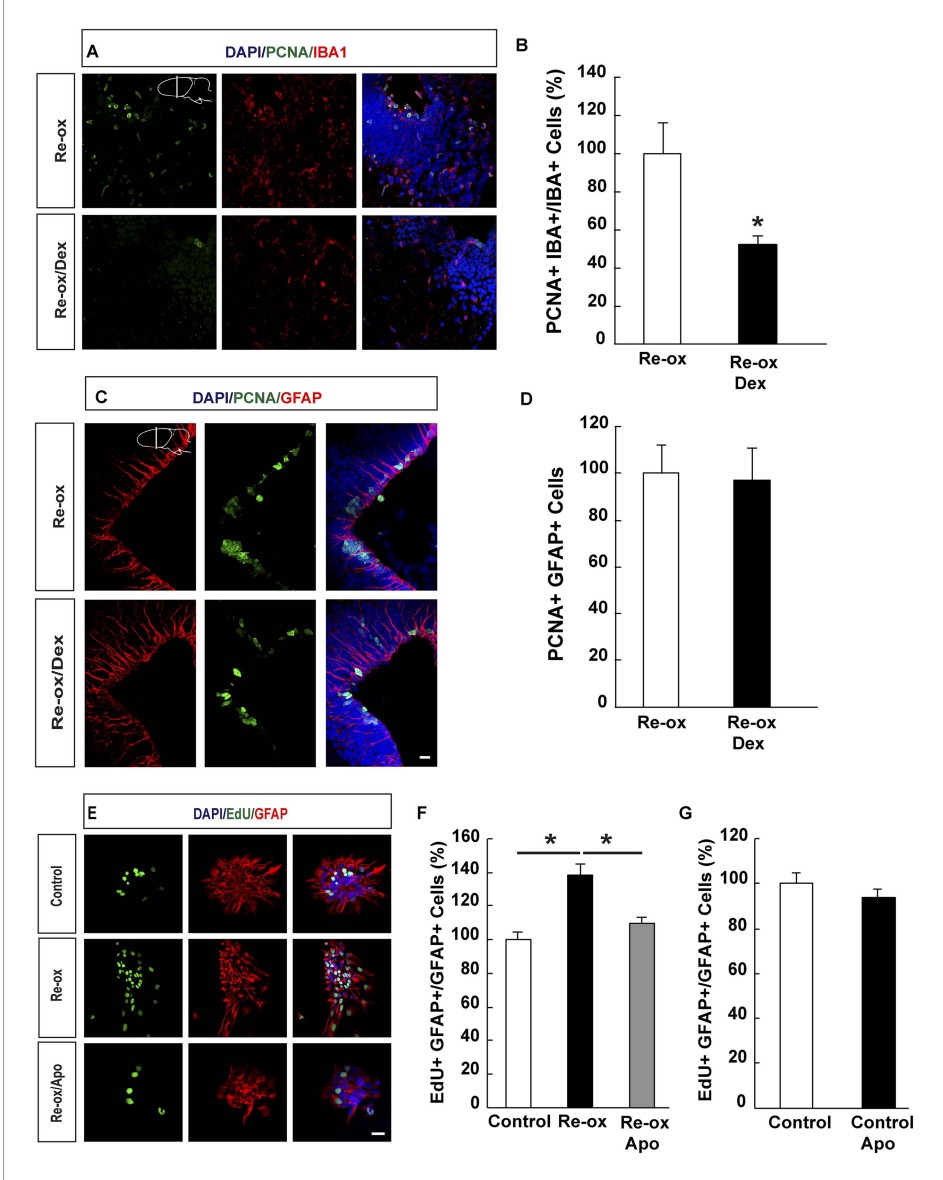

**Figure 4**. Suppression of microglia activation does not inhibit hypoxia/re-oxygenation–induced ependymoglia proliferation. (**A**, **B**) Dexamethasone inhibits microglia activation indicated by decreased number of IBA1+/PCNA+ cells. Low-magnification images illustrate that dexamethasone decreases microglia proliferation in **A** and quantification is shown in **B**. n = 6, *p < 0.05. (Unpaired t-test). (**C**, **D**) Dexamethasone does not inhibit ependymoglia activation, indicated by unchanged number of ventricular GFAP+/PCNA+ cells. Low-magnification images illustrate that dexamethasone does not decrease ependymoglia proliferation in **C** and quantification is shown in **D**. n = 6. (Unpaired t-test). (**E–G**) Hypoxia/re-oxygenation increases proliferation of GFAP+ cells in neurospheres in a ROS dependent manner. Images illustrating proliferation of GFAP+ cells in control and experimental neurosphere cultures are shown in **E**. Quantifications of PCNA+/GFAP+ cells are shown in **F**, **G**. n = 3, *p < 0.05. (Unpaired t-test). Scale bar = 20 μm.

The following source data and figure supplement are available for figure 4:

**Source data 1**. Suppression of microglia activation does not inhibit hypoxia/re-oxygenation-induced ependymoglia proliferation.

**Source data 2**. Apocynin does not inhibit microglia proliferation in vivo but abrogates neurosphere-formation after hypoxia/re-oxygenation.

*Figure 4. continued on next page*

*Figure 4. Continued*

**Figure supplement 1**. Apocynin does not inhibit microglia proliferation in vivo but abrogates neurosphere-formation after hypoxia/re-oxygenation.

manipulate ROS levels in a naturally regenerating organism. In contrast to most previous studies, which were heavily based on mixed cultures of stem and progenitor cells, we here took advantage of the fact that the vast majority of the ependymoglia cells lining the ventricles of the adult newt brain are rarely dividing stem cells (*Kirkham et al., 2014*). Our data support the view that increased ROS signaling is important for the activation of NSCs, ultimately leading to replacement of lost neurons.

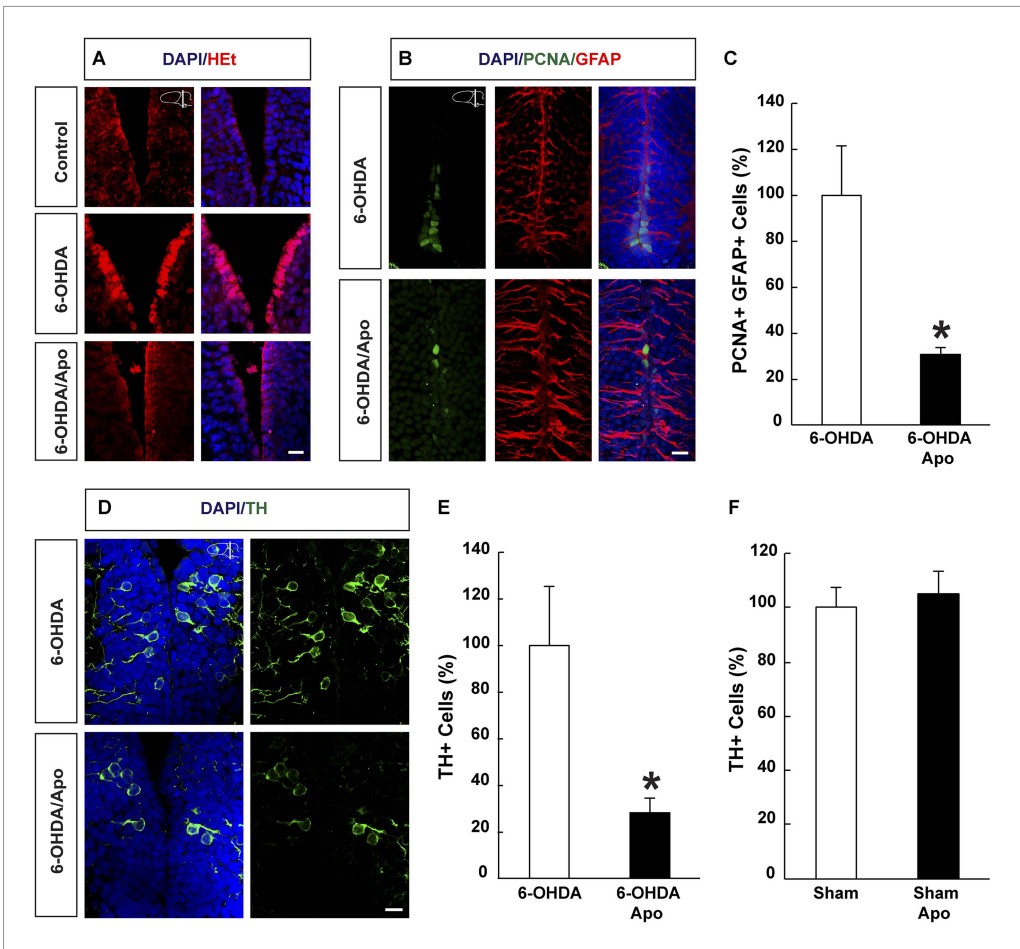

**Figure 5**. ROS dependent regeneration of midbrain dopamine neurons during normoxia. (**A**) HEt staining illustrating increased ROS levels in ependymoglia cells following ablation of dopamine neurons with 6-OHDA compared to sham-injured brains. Apocynin treatment abrogates lesion-induced increase of ROS levels. (**B**) Images illustrating that cell cycle reentry by quiescent midbrain ependymoglia after 6-OHDA-injection is inhibited by apocynin. (**C**) Quantification of cell cycle reentry by quiescent midbrain ependymoglia in the presence and absence of apocynin after 6-OHDA-injection. n = 4, *p < 0.05. (Unpaired t-test). (**D**) Images illustrating that apocynin inhibits regeneration of midbrain dopamine neuron 21 days post ablation. (**E**) Quantification of TH+ cells 21 days post ablation of dopamine neurons in the presence or absence of apocynin. n = 4–5, *p < 0.05. (Unpaired t-test). (**F**) Apocynin on its own does not change the number of TH+ cells in the midbrain in sham-ablated animals. n = 4. (Unpaired t-test). Scale bar = 20 μm.
The following source data is available for figure 5:

**Source data 1**. ROS-dependent regeneration of midbrain dopamine neurons during normoxia.

There are multiple sources of ROS in the cell and the evoked cellular response depends on the subcellular localization of ROS production. Specifically, it has been suggested that mitochondrial- and NOX-derived ROS play opposing roles in NSC and progenitor cell proliferation with mitochondria-derived ROS inhibiting and NOX-derived ROS promoting proliferation of neural progenitor proliferation (*Hou et al., 2012*; *Prozorovski et al., 2015*). In the newt brain however, mitochondrially targeted antioxidant administration neither increased nor reduced the proliferation of ependymoglia cells, although it apparently reduced mitochondrial ROS accumulation following re-oxygenation. This is in sharp contrast to the effect we found upon NOX inhibition. Since inhibiting NOX-derived ROS production counteracted re-oxygenation induced proliferation, our data indicate a dominating role for NOX-derived ROS in the control of ependymolgia cell proliferation. Other studies showed ROS accumulation in newly formed neurons, indicating a function in neuronal differentiation (*Tsatmali et al., 2006*; *Tsatmali et al., 2005*). While we could not evaluate the effect of ROS production on neuronal differentiation in vivo, our assessment did not show any increase in ROS levels in young neurons compared to NSCs in vitro. Although these observations do not demonstrate that ROS signaling is not important for differentiation, the results indicate that differentiation of neurons does not require increased ROS production.

Our experimental strategy allowed us to address a potential link between the apparent neuronal regeneration capacity of these animals and the ecological challenges within their normal habitat known to provide varying oxygen tension (*Berner and Puckett, 2010*). Completely faithful recapitulation of such constrains is not possible under laboratory conditions, not the least due to the variability of such events in nature. However, our findings are consistent with the model suggesting that shifts between hypoxic and normoxic conditions cause tissue damage (*Bickler and Buck, 2007*). We can conclude that return to physiological oxygen tension from hypoxia leads to injury response in the newt brain, shown by loss of neurons, microglia activation, cell cycle reentry by NSCs, and increased neurogenesis. Importantly, increased ROS levels are also detectable in NSCs not only after re-oxygenation but also during replacement of midbrain dopamine neurons during constantly normoxic condition. Hence, it is plausible that ROS-production in NSC has been co-opted to the capacity of replacing lost neuronal population in the newt brain. It should also be pointed out that many hypoxia tolerant vertebrates are good regenerators. Two examples for this are zebrafish and crucian carp, both of which having marked brain regeneration capacity (*Kirsche and Kirsche, 1961*; *Wolburg and Bouzehouane, 1986*; *Kroehne et al., 2011*). Zebrafish, which live in tropical waters known to become hypoxic during nighttime, is able to survive at low levels of oxygen for days and the crucian carp remains active for months in anoxic water (*Nilsson and Renshaw, 2004*; *Cao et al., 2008*; *Chu et al., 2010*). ROS signaling has also been linked to regeneration in several contexts (*Vriz et al., 2014*) and the dependence of appendage regeneration on ROS accumulation has been found in *Xenopus* larvae (*Love et al., 2013*).

The potential evolutionary relevance of this finding can also be discussed in the context of why certain animals are highly regenerative, displaying a broad spectrum of regenerative abilities in many of their tissues and body parts, while other animals are not. *Hydra*, planaria, zebrafish, and salamanders show exceptional regenerative responses as they can regrow several body parts. The seemingly random, phylogenetically uneven distribution of animals capable of regenerating multiple structures in their bodies suggests that a regeneration capacity of an organism on such scale could be a purely ancestral phenomenon, which has been lost in most species for reasons that are unclear at present (*Sanchez Alvarado, 2000*; *Simon and Tanaka, 2013*).

Nevertheless, specific micro-evolutionary selection mechanisms, such as loss of neurons during hypoxia/re-oxygenation as we demonstrate in the present work, may contribute to how an inherent regeneration capacity is manifested or whether it is manifested at all. Several recent findings support this view. First, closely related salamander species have non-overlapping range of regeneration capacities. For example, adult newts but not axolotls are able to regenerate the lens of the eye (*Grogg et al., 2005*). Similarly, some planarian species have more extensive regeneration capacities than others (*Liu et al., 2013*; *Sikes and Newmark, 2013*; *Umesono et al., 2013*). Second, we found that axolotls and newts display key cellular and molecular differences during limb regeneration (*Sandoval-Guzman et al., 2014*). Third, it has been shown that a central molecular component of salamander limb regeneration, Prod1, is only found in the salamander genome (*Garza-Garcia et al., 2010*) suggesting local evolution of limb regeneration in salamanders. Fourth, work on pectoral fin regeneration in zebrafish revealed an intriguing example of a sex-specific obstruction of regeneration,

likely due to interference with a signaling pathway maintaining key secondary sexual attributes (*Kang et al., 2013*).

In a cross-species comparative setting, it is also noteworthy that cell cycle reentry by NSCs appears to be dependent on accumulation of inflammatory cells and microglia activation in the zebrafish brain (*Kyritsis et al., 2012*). The hypoxia/re-oxygenation experimental paradigm that we employed does not provide evidence for such an interaction in the newt brain and our data rather suggest a cell autonomous process. Although the experimental manipulations used in zebrafish and in the newt brain here are different from each other, and we cannot rule out the possibility that microglia could activate NSCs in the newt brain under certain conditions, our observations indicate that the two animal species embark on at least partially non-overlapping signaling mechanism during neuronal regeneration.

While the systemic manipulation of oxygen tension and ROS accumulation led to increased neuronal death, we could not detect NSCs with apoptotic phenotype. This difference indicates that NSCs resist to ROS related damages. Future studies should address the identity of the molecular programs underlying NSCs survival and cell cycle re-activation as a response to increased ROS levels in the brain.

## Materials and methods

### Animals

All experiments were performed on adult red-spotted newts, *Notophthalmus viridescens* (Charles Sullivan, Nashville, TN, USA) according to European Community and local ethics committee guidelines.

### Experimental manipulation of oxygen tension

Newts were placed in an aquarium sealed with plastic lid. Nitrogen gas was perfused into the water via an air diffuser to make the environment hypoxic. Gas flow was regulated by a valve, which was controlled by an $O_2$-sensor in the aquarium via an electrode. $O_2$ tension was gradually reduced during 48 hr to finally reach 10% of normal level and subsequently brought back to normoxia as indicated in *Figure 1—figure supplement 1*.

### Visualization of ROS

Brains were dissected out and incubated in 100 µM HEt (Thermo Fisher Scientific, Waltham, MA) or 10 µM Mitosox (Thermo Fisher Scientific, Waltham, MA) solution for 5 to 15 min in a dark chamber at room temperature. Then, they were fixed in 4% formaldehyde and sectioned. Alternatively, HEt 10 mg/kg was injected intravenously and the animals were left sedated for 1 hr. Animals were then perfused and brains were isolated and sectioned.

### Administration of substances

Apocynin (Sigma, 5 mg/kg) was injected intraperitoneally immediately after hypoxia twice per day for 3 days.

Mitotempo (Sigma, 5 mg/kg) was injected intraperitoneally immediately after hypoxia twice per day for 3 days.

Dexamethasone (Sigma, 2 mg/kg) was injected intraperitoneally twice per day for 5 days before newts were shifted to hypoxia and for 3 days immediately after hypoxia.

EdU (Invitrogen, Carlsbad, CA, 50 mg/kg) was injected intraperitoneally twice per day during reperfusion, from day 4 till day 8 and animals chased for 35 days for assessing neuronal differentiation. For assessing cell proliferation, EdU (50 mg/kg) was injected intraperitoneally 2 hr before sacrifice.

6-OHDA was injected intracranially as described earlier (*Berg et al., 2010*). During dopamine neuron regeneration experiments, apocynin (5 mg/kg) was administered between day 4 and day 9 after 6-OHDA-injection.

### Immunochemistry

Newts were sedated with 0.1% Tricane (Sigma, St. Louis, MO) solution and perfused with 4% formaldehyde and cryo-protected in sucrose at 4°C overnight. 20-µm serial coronal sections were

made alternating on five slides. Sections were post-fixed with 4% formaldehyde solution for five minutes followed by 3 × 5 minutes wash in PBS. Sections were treated with 0.1% Triton X-100 in PBS (Sigma) for 15 min at RT. For PCNA staining, sections were incubated with 2M HCl in 0.5% Triton X-100 in PBS for 20 min at 37°C and washed 3 × 3 minutes with PBS. All sections were blocked in blocking solutions, containing 5% donkey serum, 0.5% Triton X-100 in PBS for 30 min at RT. Subsequently, sections were incubated with one of the following primary antibodies in blocking solutions overnight: mouse anti-PCNA (1:500; Millipore, Temecula, CA), rabbit anti-Mitofusin-1 (1:500; Cell signaling, Danvers, MA), goat anti-DCX (1:500; Santacruz, Paso Robles, CA), rabbit anti-IBA1 (1:500; Wako, Richmond, VA), rabbit anti-TH (1:500; Millipore), mouse anti-NeuN (1:500; Millipore), and mouse anti-HuC/HuD (1:500; Millipore). Next day, sections were washed 3 × 5 minutes in PBS, and incubated with the added appropriate secondary antibody (1:500; Molecular probes, Eugene, OR) in blocking solutions for 2 hr at room temperature. EdU staining was performed by incubating sections with 100 mM Tris, 1 mM $CuSO_4$, 50-100 µM fluorescent azide, and 100 mM ascorbic acid as prescribed in (*Salic and Mitchison, 2008*). TUNEL staining was performed according to the manufacturer's protocol (Roche).

## Primary cell culture

Primary cell culture was performed as previously described (*Kirkham et al., 2014*). Isolated cells were plated in 25 cm² flasks and left at 25°C with 2% CO2. After 24 hr, the cells were shifted to 1% oxygen for 24 hr. Subsequently, cells were shifted to normoxia and left for 2 weeks. Fresh medium was added every fourth day and the number of neurospheres was assessed after 14 days. For blocking ROS production, apocynin was added to a final concentration of 100 µM immediately after hypoxia for 3 days. For proliferation assay, EdU was added to neurosphere cultures to a final concentration of 20 µM and spheres were transferred to Poly-D-Lysine-coated slides. To measure ROS intensity, live cells were incubated with 30 µM HEt solution for 30 min at RT. Cells were washed with L15 medium and immediately fixed with 4% PFA, proceeded for immunocytochemistry.

## Image analysis and counting

The number of positive cells was quantified under 20x magnification with the optical fractionator method on systemic random sampling of every fifth sections along the rostro–caudal axis. Images for cell counting were captured with LSM-700 using ZEN software. To analyze the double-labeled cells, 20x confocal images along the entire Z-axis using 1-µm intervals were taken and counted. For each animal, totally 10 sections in forebrain for hypoxia/re-oxygenation studies and 5 sections in midbrain for regeneration studies following 6-OHDA-injection were analyzed. To assess ROS signal in cultured cells, HEt fluorescence intensity was measured in the cell nuclei and averaged from the data of 10–25 cells/biological sample. Images were processed with either Photoshop (Adobe) or with Image J using linear adjustments.

## Statistical analyses

Animals were randomly chosen for each experiment and allocated into groups in a non-biased manner. Normal distribution of sample data was determined by using the Shapiro–Wilk test. We performed unpaired two-tailed t-tests for samples that were normally distributed, and Mann–Whitney test performed for samples that were not normally distributed. Sample size (n) is indicated in each experiment. Error bars represent SEM. Results are considered statistically significant at $p < 0.05$.

## Acknowledgements

We thank A Elewa, N Dantuma, C Sjögren for many helpful comments on the manuscript, and H Wang and M Kirkham for advice. This work was supported by grants from the European Research Council, Swedish Research Council, Swedish Cancer Society, AFA Insurances to AS. YC´s laboratory is supported by research grants from the Swedish Research Council, the Swedish Cancer Foundation, the Karolinska Institute Foundation, the Karolinska Institute distinguished professor award, the Torsten Soderbergs foundation, the NOVO Nordisk Foundation, the Advanced grant from the NOVO Nordisk foundation, and the Alice Wallenberg foundation.

## Additional information

### Funding

| Funder | Author |
|---|---|
| European Research Council (ERC) | András Simon |
| AFA Försäkring | András Simon |
| Vetenskapsrådet | Yihai Cao, András Simon |
| Cancerfonden | András Simon |
| Torsten Söderbergs Stiftelse | Yihai Cao |
| Novo Nordisk | Yihai Cao |

The funders had no role in study design, data collection and interpretation, or the decision to submit the work for publication.

### Author contributions

LSH, DAB, Conception and design, Acquisition of data, Analysis and interpretation of data, Drafting or revising the article; LB, Analysis and interpretation of data, Drafting or revising the article; LDJ, Analysis and interpretation of data, Contributed unpublished essential data or reagents; YC, Drafting or revising the article, Contributed unpublished essential data or reagents; AS, Conception and design, Analysis and interpretation of data, Drafting or revising the article

### Ethics

Animal experimentation: The protocols were performed in accordance with EU regulations and were approved by local ethics committee (Permission number N429/12).

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
