## [Decision Letter]

Thank you for submitting your work entitled “Environmental changes in oxygen tension reveal ROS-dependent neurogenesis and regeneration in the adult newt brain” for peer review at *eLife*. Your submission has been favorably evaluated by Janet Rossant (Senior Editor), Alejandro Sánchez Alvarado (Reviewing Editor), and two reviewers.

The reviewers have discussed the reviews with one another and the Reviewing Editor has drafted this decision to help you prepare a revised submission.

In this manuscript, Hameed et al. demonstrate the importance of ROS in neuronal regeneration in the red spotted newt. The authors first show that neuronal death and ependymoglia proliferation occur during the reoxigenation phase during hypoxia and then proceed to demonstrate that ROS levels increase in the ependymoglia during this time. More importantly, the authors show that by interfering with ROS activity, they can experimentally and significantly reduce both ependymoglia proliferation and the ensuing neurogenesis to replace dead neurons. Perhaps the most exciting aspect of the work presented in the manuscript is the evidence that ROS levels are also elevated in the 6OHDA model killing dopaminergic neurons and required also here for regenerative neurogenesis. In sum, the work reveals an important role for ROS levels during regenerative neurogenesis with will likely be of great significance in studies of regeneration of various organs in other organisms, including mammals.

Before this interesting article can be considered for publication, the Reviewing Editor and the reviewers would like the authors to consider and address the following essential issues:

1) A central issue of the present manuscript is that we do not know what the molecular consequence of ROS are in the experimental context, i.e. the down-stream mechanisms. ROS has been involved in neurogenesis (see review by [32]) and depending on the precise localization within the cell, different effects are mediated: NSC self-renewal via a NOX-mediated pathway; or neuronal differentiation via mitochondrial ROS increase. It would be desirable, therefore, to see if mechanistic insight can be derived from the experimental paradigm reported by Hameed et al., specifically whether blockers with further specificity of the NOX- or mitochondrial pathways alter the reported outcomes. Alternatively, it would also be interesting to determine the duration of the ROS response and the time at which adding the apocynin blocker has no longer an effect. For example in the 6OHD model, the authors start giving apocynin at 4 days after the lesion – what about starting after 7 days – would this have no effect anymore? Do young generated neurons also have elevated ROS levels? This could help explaining whether ROS is relevant only to initiate the proliferative response but no longer required for later neuronal differentiation.

2) The authors need to refine the focus of their manuscript. Given the emphasis on mechanism in the Introduction and Results sections, a more in-depth treatment of the findings and results in the Discussion section seems appropriate. Presently, the Discussion focuses primarily on evolutionary considerations, and while these are important, they should be a part, but not the central content of the Discussion section.

3) The stringency of data analyses needs to be increased. For example, the reviewers would like to better understand the statistics presented. It is understood that one “*n”* is one animal, but how many sections/cells were counted? It is also understood that the authors used a *t*-test, but did the authors tested whether the data follow a Gaussian normal distribution? If this was not the case, then the *t*-test is not the appropriate statistical analysis method.

4) The authors should be more consistent in the use of cell proliferation assays in order to facilitate comparisons between experiments within the manuscript. In some conditions the authors chose direct incorporation of nucleotides as a clear measure of cell-cycle reentry, while in others they use PCNA immunostaining. Because PCNA positive cells can be detected in the absence of S-phase reentry, comparison between the two types of experimental sets is complicated. Given the comparative nature of the experiments performed, it seems necessary to use one consistent assay, preferentially EdU incorporation.

---

## [Author Response]

Before this interesting article can be considered for publication, the Reviewing Editor and the reviewers would like the authors to consider and address the following essential issues:

*1) A central issue of the present manuscript is that we do not know what the molecular consequence of ROS are in the experimental context, i.e., the down-stream mechanisms. ROS has been involved in neurogenesis (see review by*
[32]*) and depending on the precise localization within the cell, different effects are mediated: NSC self-renewal via a NOX-mediated pathway; or neuronal differentiation via mitochondrial ROS increase. It would be desirable, therefore, to see if mechanistic insight can be derived from the experimental paradigm reported by Hameed et al., specifically whether blockers with further specificity of the NOX- or mitochondrial pathways alter the reported outcomes. Alternatively, it would also be interesting to determine the duration of the ROS response and the time at which adding the apocynin blocker has no longer an effect. For example in the 6OHD model, the authors start giving apocynin at 4 days after the lesion – what about starting after 7 days – would this have no effect anymore? Do young generated neurons also have elevated ROS levels? This could help explaining whether ROS is relevant only to initiate the proliferative response but no longer required for later neuronal differentiation.*

We performed several experiments to address the specific questions.

Subcellular localization of ROS production: We first determined whether mitochondrial ROS increased during re-oxygenation. To detect mitochondrial ROS we injected animals with the mitochondrial superoxide indicator Mitosox and could conclude that animals that underwent hypoxia/re-oxygenation displayed accumulation of mitochondrial ROS compared to control animals. This accumulation could be blocked by administration of the mitochondrially targeted antioxidant, Mitotempo. However, in contrast to the effect of NOX-inhibition with apocynin, Mitotempo administration did not reduce the number of PCNA^+ ^ependymoglia cells. Thus our data indicate a dominating role for NOX derived ROS in the control of ependymoglia cell proliferation. These new data are shown in Figure 3 and are referred to in the Results, subsection “Ependymoglia cell proliferation and neurogenesis following re-oxygenation depends on ROS production”.

ROS and differentiation: To address whether ROS impinges on differentiation of young neurons in the newt brain, we carried out pulse chase experiments with the nucleotide analogue EdU in combination with subsequent inhibition of mitochondrial and NOX derived ROS. Unfortunately most animals did not tolerate the number of sequential injections required for this study and hence these experiments could not be evaluated. The suggested alternative way to address this question using the 6-OHDA mediated ablation model is unfortunately not feasible because the proliferation and differentiation phases are not temporally distinct during regeneration. Differentiation of TH neurons occurs gradually over time rather than in a separate burst after an initial proliferation phase. This precludes us to disentangle differentiation and proliferation from each other in a temporal dimension. However, as an indirect way we measured ROS levels in GFAP^+^ NSCs and Dcx^+^ young neurons and did not detect a difference comparing the two cell types. Although these observations do not demonstrate that ROS signaling is not important for differentiation, the results indicate that differentiation of neurons does not require increased ROS production. The data are shown in Figure 3—figure supplement 1 and described in the Results, subsection “Ependymoglia cell proliferation and neurogenesis following re-oxygenation depends on ROS production”.

2) The authors need to refine the focus of their manuscript. Given the emphasis on mechanism in the Introduction and Results sections, a more in-depth treatment of the findings and results in the Discussion section seems appropriate. Presently, the Discussion focuses primarily on evolutionary considerations, and while these are important, they should be a part, but not the central content of the Discussion section.

We complemented the Discussion according to this request. In the revised version the first part of the Discussion entirely focuses on mechanisms of ROS generation. The second part of the Discussion addresses the hypotheses put forward in the Introduction, namely that animals need to repair tissues that become damaged during hypoxia/re-oxygenation, and that hypoxia/re-oxygenation leads to events in the newt brain that share common features with processes taking place after experimental ablation and subsequent regeneration of neurons during normoxia.

*3) The stringency of data analyses needs to be increased. For example, the reviewers would like to better understand the statistics presented. It is understood that one “*n” *is one animal, but how many sections/cells were counted? It is also understood that the authors used a* t*-test, but did the authors tested whether the data follow a Gaussian normal distribution? If this was not the case, then the* t*-test is not the appropriate statistical analysis method.*

We incorporated a description of the statistical analyses including counting procedures and determination of normal distribution (see Methods, subsections “Image Analysis and counting” and “Statistical analyses”). In addition we indicate now in each figure legend which statistical test was used.

4) The authors should be more consistent in the use of cell proliferation assays in order to facilitate comparisons between experiments within the manuscript. In some conditions the authors chose direct incorporation of nucleotides as a clear measure of cell-cycle reentry, while in others they use PCNA immunostaining. Because PCNA positive cells can be detected in the absence of S-phase reentry, comparison between the two types of experimental sets is complicated. Given the comparative nature of the experiments performed, it seems necessary to use one consistent assay, preferentially EdU incorporation.

In the in vivo assays ependymoglia proliferation were consistently determined by PCNA staining and only differentiation was determined in EdU pulse/chase studies (Figure 2). We have previously shown ([2]; Development) that PCNA staining and EdU pulsing give with each other consistent results regarding ependymoglia proliferation. In addition, as a proof of principle in the context of the present study, we performed EdU pulse experiments during re-oxygenation and compared the results with data obtained from PCNA staining. Importantly, there was no difference in the conclusions from the two types of measurements, as both PCNA staining and EdU pulsing indicated a 1.8 increase in ependymoglia proliferation upon re-oxygenation compared to control animals. These data are presented in the Figure 2—figure supplement 1 and are referred to in Results (in the subsection “Re-oxygenation leads to cell cycle reentry by forebrain ependymoglia cells and increased neurogenesis”).